# ABDUCTIVE COMMONSENSE REASONING

**Chandra Bhagavatula**◇**, Ronan Le Bras**◇**, Chaitanya Malaviya**◇**, Keisuke Sakaguchi**◇**,**
**Ari Holtzman**◇**, Hannah Rashkin**◇**, Doug Downey**◇**, Scott Wen-tau Yih**♣**, Yejin Choi**◇♡
◇Allen Institute for AI, Seattle, WA, USA, ♣Facebook AI, Seattle, WA, USA
♡Paul G. Allen School of Computer Science & Engineering, WA, USA
{chandrab,ronanlb,chaitanyam,keisukes}@allenai.org
{arih,hannahr,dougd}@allenai.org
{yejin}@cs.washington.edu
{scottyih}@fb.com*

## ABSTRACT

*Abductive reasoning* is inference to the *most plausible explanation*. For example, if Jenny finds her house in a mess when she returns from work, and remembers that she left a window open, she can hypothesize that a thief broke into her house and caused the mess, as the most plausible explanation. While abduction has long been considered to be at the core of how people interpret and read between the lines in natural language (Hobbs et al., 1988), there has been relatively little research in support of abductive natural language inference and generation.

We present the first study that investigates the viability of language-based abductive reasoning. We introduce a challenge dataset, *ART*, that consists of over 20k commonsense narrative contexts and 200k explanations. Based on this dataset, we conceptualize two new tasks – (i) *Abductive NLI*: a multiple-choice question answering task for choosing the more likely explanation, and (ii) *Abductive NLG*: a conditional generation task for explaining given observations in natural language. On *Abductive NLI*, the best model achieves 68.9% accuracy, well below human performance of 91.4%. On *Abductive NLG*, the current best language generators struggle even more, as they lack reasoning capabilities that are trivial for humans. Our analysis leads to new insights into the types of reasoning that deep pre-trained language models fail to perform—despite their strong performance on the related but more narrowly defined task of *entailment* NLI—pointing to interesting avenues for future research.

## 1 INTRODUCTION

> *The brain is an abduction machine, continuously trying to prove abductively that the observables in its environment constitute a coherent situation.*
> *– Jerry Hobbs, ACL 2013 Lifetime Achievement Award[1]*

*Abductive reasoning* is inference to the most plausible explanation for incomplete observations (Peirce, 1965a). Figure 1 illustrates an example. Given the incomplete observations about the world that $O_1$: "Jenny cleaned her house and went to work, leaving the window just a crack open." and sometime later $O_2$: "When Jenny returned home, she saw her house was a mess.", we can hypothesize different potential explanations and reason about which is the most likely. We can readily rule out $H_3$ since it fails to justify the observation $O_2$. While $H_1$ and $H_2$ are both plausible, the most likely explanation based on commonsense is $H_1$ as $H_2$ is somewhat implausible given $O_1$.

One crucial observation Peirce makes about abductive reasoning is that abduction is *"the only logical operation which introduces any new ideas"*, which contrasts with other types of inference such as entailment, that focuses on inferring only such information that is already provided in the premise.

---

*Work done while at AI2

[1]The full transcript of his award speech is available at https://www.mitpressjournals.org/doi/full/10.1162/COLI_a_00171

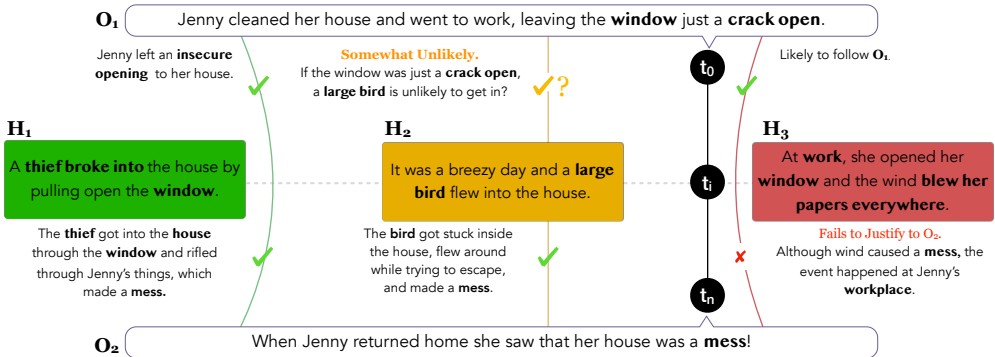

Figure 1: Example of Abductive Reasoning. Given observations $O_1$ and $O_2$, the $\alpha$NLI task is to select the most plausible explanatory hypothesis. Since the number of hypotheses is massive in any given situation, we make a simplifying assumption in our $\mathcal{ART}$ dataset to only choose between a pair of explanations.

Abductive reasoning has long been considered to be at the core of understanding narratives (Hobbs et al., 1988), reading between the lines (Norvig, 1987; Charniak & Shimony, 1990), reasoning about everyday situations (Peirce, 1965b; Andersen, 1973), and counterfactual reasoning (Pearl, 2002; Pearl & Mackenzie, 2018). Despite the broad recognition of its importance, however, the study of abductive reasoning in narrative text has very rarely appeared in the NLP literature, in large part because most previous work on abductive reasoning has focused on formal logic, which has proven to be too rigid to generalize to the full complexity of natural language.

In this paper, we present the first study to investigate the viability of language-based abductive reasoning. This shift from *logic*-based to *language*-based reasoning draws inspirations from a significant body of work on language-based entailment (Bowman et al., 2015; Williams et al., 2018b), language-based logic (Lakoff, 1970; MacCartney & Manning, 2007), and language-based commonsense reasoning (Mostafazadeh et al., 2016; Zellers et al., 2018). In particular, we investigate the use of natural language as the representation medium, and probe deep neural models on language-based abductive reasoning.

More concretely, we propose Abductive Natural Language Inference ($\alpha$NLI) and Abductive Natural Language Generation ($\alpha$NLG) as two novel reasoning tasks in narrative contexts.[2] We formulate $\alpha$NLI as a multiple-choice task to support easy and reliable automatic evaluation: given a context, the task is to choose the more likely explanation from a given pair of hypotheses choices. We also introduce a new challenge dataset, $\mathcal{ART}$, that consists of 20K narratives accompanied by over 200K explanatory hypothesis.[34] We then establish comprehensive baseline performance based on state-of-the-art NLI and language models. The best baseline for $\alpha$NLI based on BERT achieves 68.9% accuracy, with a considerable gap compared to human performance of 91.4%(§5.2). The best generative model, based on GPT2, performs well below human performance on the $\alpha$NLG task (§5.2). Our analysis leads to insights into the types of reasoning that deep pre-trained language models fail to perform — despite their strong performance on the closely related but different task of *entailment* NLI — pointing to future research directions.

## 2    TASK DEFINITION

**Abductive Natural Language Inference**    We formulate $\alpha$NLI as multiple choice problems consisting of a pair of observations as context and a pair of hypothesis choices. Each instance in $\mathcal{ART}$ is defined as follows:

- $O_1$: The observation at time $t_1$.

---

[2]$\alpha$NLI and $\alpha$NLG are pronounced as *alpha*-NLI and *alpha*-NLG, respectively

[3]$\mathcal{ART}$: **A**bductive **R**easoning in narrative **T**ext.

[4]Data available to download at http://abductivecommonsense.xyz

- $O_2$: The observation at time $t_2 > t_1$.
- $h^+$: A *plausible* hypothesis that explains the two observations $O_1$ and $O_2$.
- $h^-$: An *implausible* (or *less* plausible) hypothesis for observations $O_1$ and $O_2$.

Given the observations and a pair of hypotheses, the $\alpha$NLI task is to select the most plausible explanation (hypothesis).

**Abductive Natural Language Generation**  $\alpha$NLG is the task of generating a valid hypothesis $h^+$ given the two observations $O_1$ and $O_2$. Formally, the task requires to maximize $P(h^+|O_1, O_2)$.

## 3  MODELS FOR ABDUCTIVE COMMONSENSE REASONING

### 3.1  ABDUCTIVE NATURAL LANGUAGE INFERENCE

**A Probabilistic Framework for $\alpha$NLI:**  A distinct feature of the $\alpha$NLI task is that it requires jointly considering all available observations and their commonsense implications, to identify the correct hypothesis. Formally, the $\alpha$NLI task is to select the hypothesis $h^*$ that is most probable given the observations.

$$h^* = \arg\max_{h^i} P(H = h^i | O_1, O_2) \tag{1}$$

Rewriting the objective using Bayes Rule conditioned on $O_1$, we have:

$$P(h^i|O_1, O_2) \propto P(O_2|h^i, O_1)P(h^i|O_1) \tag{2}$$

We formulate a set of probabilistic models for $\alpha$NLI that make various independence assumptions on Equation 2 – starting from a simple baseline that ignores the observations entirely, and building up to a fully joint model. These models are depicted as Bayesian Networks in Figure 2.

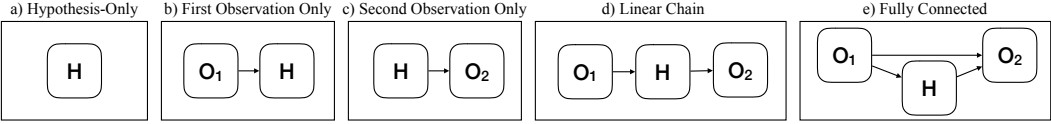

Figure 2: Illustration of the graphical models described in the probabilistic framework. The "Fully Connected" model can, in theory, combine information from both available observations.

**Hypothesis Only:**  Our simplest model makes the strong assumption that the hypothesis is entirely independent of both observations, i.e. $(H \perp O_1, O_2)$, in which case we simply aim to maximize the marginal $P(H)$.

**First (or Second) Observation Only:**  Our next two models make weaker assumptions: that the hypothesis depends on only one of the first $O_1$ or second $O_2$ observation.

**Linear Chain:**  Our next model uses both observations, but considers each observation's influence on the hypothesis *independently*, i.e. it does not combine information across the observations. Formally, the model assumes that the three variables $\langle O_1, H, O_2 \rangle$ form a linear Markov chain, where the second observation is conditionally independent of the first, given the hypothesis (i.e. $(O_1 \perp O_2|H)$). Under this assumption, we aim to maximize a somewhat simpler objective than Equation 2:

$$h^* = \arg\max_{h^i} P(O_2|h^i)P(h^i|O_1) \text{ where } (O_1 \perp O_2|H) \tag{3}$$

**Fully Connected:**  Finally, our most sophisticated model jointly models all three random variables as in Equation 2, and can in principle combine information across both observations to choose the correct hypothesis.

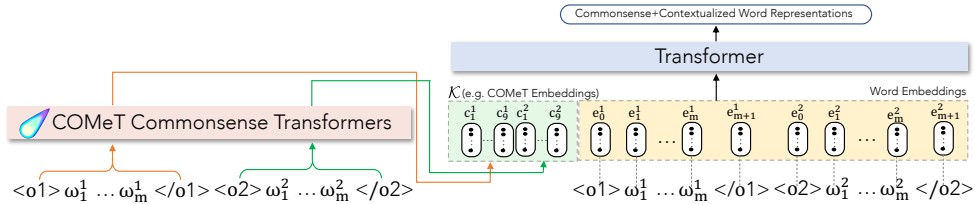

Figure 3: Overview of an $\alpha$NLG model that integrates commonsense representations obtained from COMeT (Bosselut et al., 2019) with GPT2. Each observation is input to the COMeT model to obtain nine embeddings, each associated with one commonsense inference type.

To help illustrate the subtle distinction between how the Linear Chain and Fully Connected models consider both observations, consider the following example. Let observation $O_1$: "Carl went to the store desperately searching for flour tortillas for a recipe." and $O_2$: "Carl left the store very frustrated.". Then consider two distinct hypotheses, an incorrect $h^1$: "The cashier was rude" and the correct $h^2$: "The store had corn tortillas, but not flour ones.". For this example, a Linear Chain model could arrive at the wrong answer, because it reasons about the observations separately—taking $O_1$ in isolation, both $h^1$ and $h^2$ seem plausible next events, albeit each *a priori* unlikely. And for $O_2$ in isolation—i.e. in the absence of $O_1$, as for a randomly drawn shopper—the $h^1$ explanation of a rude cashier seems a much more plausible explanation of Carl's frustration than are the details of the store's tortilla selection. Combining these two separate factors leads the Linear Chain to select $h^1$ as the more plausible explanation. It is only by reasoning about Carl's goal in $O_1$ *jointly* with his frustration in $O_2$, as in the Fully Connected model, that we arrive at the correct answer $h^2$ as the more plausible explanation.

In our experiments, we encode the different independence assumptions in the best performing neural network model. For the hypothesis-only and single observation models, we can enforce the independencies by simply restricting the inputs of the model to only the relevant variables. On the other hand, the Linear Chain model takes all three variables as input, but we restrict the form of the model to enforce the conditional independence. Specifically, we learn a discriminative classifier:

$$P_{\text{Linear Chain}}(h|O_1, O_2) \propto e^{\phi(O_1,h)+\phi'(h,O_2)}$$

where $\phi$ and $\phi'$ are neural networks that produce scalar values.

## 3.2 ABDUCTIVE NATURAL LANGUAGE GENERATION

Given $h^+ = \{w_1^h \ldots w_l^h\}$, $O_1=\{w_1^{o1} \ldots w_m^{o1}\}$ and $O_2=\{w_1^{o2} \ldots w_n^{o2}\}$ as sequences of tokens, the $\alpha$NLG task can be modeled as $P(h^+|O_1, O_2) = \prod P(w_i^h|w_{<i}^h, w_1^{o1} \ldots w_m^{o1}, w_1^{o2} \ldots w_n^{o2})$ Optionally, the model can also be conditioned on background knowledge $\mathcal{K}$. Parameterized models can then be trained to minimize the negative log-likelihood over instances in $\mathcal{ART}$:

$$\mathcal{L} = -\sum_{i=1}^{N} \log P(w_i^h|w_{<i}^h, w_1^{o1} \ldots w_m^{o1}, w_1^{o2} \ldots w_n^{o2}, \mathcal{K}) \tag{4}$$

## 4 $\mathcal{ART}$ DATASET: ABDUCTIVE REASONING IN NARRATIVE TEXT

$\mathcal{ART}$ is the first large-scale benchmark dataset for studying abductive reasoning in narrative texts. It consists of $\sim$20K narrative contexts (pairs of observations $\langle O_1, O_2 \rangle$) with over 200K explanatory hypotheses. Table 6 in the Appendix summarizes corpus-level statistics of the $\mathcal{ART}$ dataset.[5] Figure 4 shows some illustrative examples from $\mathcal{ART}$ (dev split). The best model based on BERT fails to correctly predict the first two dev examples.

---

[5]We will publicly release the $\mathcal{ART}$ dataset upon acceptance.

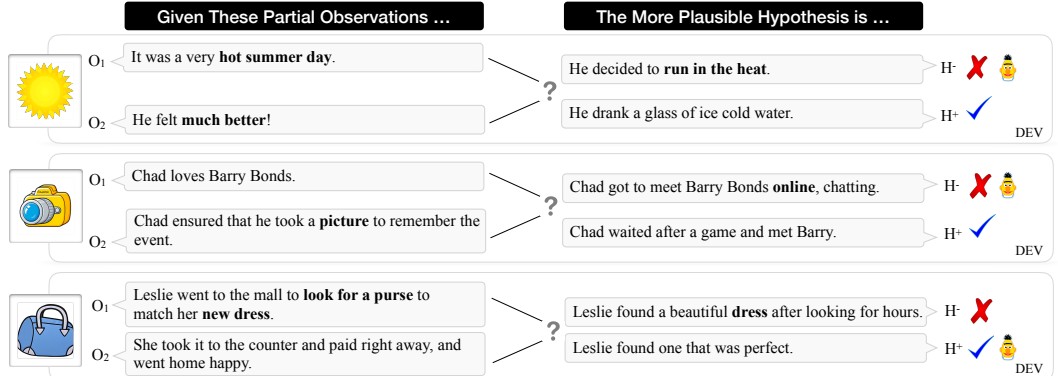

Figure 4: Examples from 𝒜ℛ𝒯 (dev split). The best model based on BERT fails to correctly predict the first two examples.

**Collecting Observations:** The pairs $O_1$, $O_2$ in 𝒜ℛ𝒯 are drawn from the ROCStories dataset (Mostafazadeh et al., 2016). ROCStories is a large collection of short, manually curated five-sentence stories. It was designed to have a clear beginning and ending for each story, which naturally map to the first ($O_1$) and second ($O_2$) observations in 𝒜ℛ𝒯.

**Collecting Hypotheses Options:** We crowdsourced the plausible and implausible hypotheses options on Amazon Mechanical Turk (AMT) in two separate tasks[6]:

1. Plausible Hypothesis Options: We presented $O_1$ and $O_2$ as narrative context to crowdworkers who were prompted to fill in "What happened in-between?" in natural language. The design of the task motivates the use of abductive reasoning to hypothesize likely explanations for the two given observations.

2. Implausible Hypothesis Options: In this task, we presented workers with observations $O_1, O_2$ and one plausible hypothesis option $h^+ \in \mathcal{H}^+$ collected from the previous task. Crowdworkers were instructed to make minimal edits (up to 5 words) to a given $h^+$ to create implausible hypothesis variations for each plausible hypothesis.

A significant challenge in creating datasets is avoiding *annotation artifacts* – unintentional patterns in the data that leak information about the target label – that several recent studies (Gururangan et al., 2018; Poliak et al., 2018; Tsuchiya, 2018) have reported on crowdsourced datasets . To tackle this challenge, we collect multiple plausible and implausible hypotheses for each $\langle O_1, O_2 \rangle$ pair (as described above) and then apply an adversarial filtering algorithm to retain one challenging pair of hypotheses that are hard to distinguish between. We describe our algorithm in detail in Appendix A.5. While our final dataset uses BERT as the adversary, preliminary experiments that used GPT as an adversary resulted in similar drops in performance of all models, including all BERT variants. We compare the results of the two adversaries in Table 1.

## 5 EXPERIMENTS AND RESULTS

We now present our evaluation of finetuned state-of-the-art pre-trained language models on the 𝒜ℛ𝒯 dataset, and several other baseline systems for both αNLI and αNLG. Since αNLI is framed as a binary classification problem, we choose *accuracy* as our primary metric. For αNLG, we report performance on automated metrics such as BLEU (Papineni et al., 2002), CIDEr (Vedantam et al., 2015), METEOR (Banerjee & Lavie, 2005) and also report human evaluation results.

### 5.1 ABDUCTIVE NATURAL LANGUAGE INFERENCE

---

[6]Both crowdsourcing tasks are complex and require creative writing. Along with the 𝒜ℛ𝒯 dataset, we will publicly release templates and the full set of instructions for all crowdsourcing tasks to facilitate future data collection and research in this direction.

Despite strong performance on several other NLP benchmark datasets, the best baseline model based on BERT achieves an accuracy of just 68.9% on $\mathcal{ART}$ compared to human performance of 91.4%. The large gap between human performance and that of the best system provides significant scope for development of more sophisticated abductive reasoning models. Our experiments show that introducing the additional independence assumptions described in Section 3.1 over the fully connected model tends to degrade system performance (see Table 1) in general.

| Model | GPT AF Acc. (%) | $\mathcal{ART}$ Acc. (%) |
|---|---|---|
| Random (2-way choice) | 50.1 | 50.4 |
| Majority (from *dev* set) | 50.1 | 50.8 |
| Infersent (Conneau et al., 2017) | 50.9 | 50.8 |
| ESIM+ELMo (Chen et al., 2017) | 58.2 | 58.8 |
| **Finetuning Pre-trained LMs** | | |
| GPT-ft | 52.6 (0.9) | 63.1 (0.5) |
| BERT-ft [$h^i$ Only] | 55.9 (0.7) | 59.5 (0.2) |
| BERT-ft [$O_1$ Only] | 63.9 (0.8) | 63.5 (0.7) |
| BERT-ft [$O_2$ Only] | 68.1 (0.6) | 66.6 (0.2) |
| BERT-ft [Linear Chain] | 65.3 (1.4) | 68.9 (0.5) |
| BERT-ft [Fully Connected] | 72.0 (0.5) | 68.6 (0.5) |
| **Human Performance** | - | 91.4 |

Table 1: Performance of baselines and finetuned-LM approaches on the *test* set of $\mathcal{ART}$. Test accuracy is reported as the mean of five models trained with random seeds, with the standard deviation in parenthesis.

**Human Performance**    We compute human performance using AMT. Each instance (two observations and two hypothesis choices) is shown to three workers who were prompted to choose the more plausible hypothesis choice.[7] We compute majority vote on the labels assigned which leads to a human accuracy of 91.4% on the $\mathcal{ART}$ test set.

**Baselines**    We include baselines that rely on simple features to verify that $\mathcal{ART}$ is not trivially solvable due to noticeable annotation artifacts, observed in several crowdsourced datasets. The accuracies of all simple baselines are close to chance-performance on the task – indicating that the dataset is free of simple annotation artifacts.

A model for the related but distinct task of entailment NLI (e.g. SNLI) forms a natural baseline for $\alpha$NLI. We re-train the ESIM+ELMo (Chen et al., 2017; Peters et al., 2018) model as its performance on entailment NLI (88.9%) is close to state-of-the-art models (excluding pre-trained language models). This model only achieves an accuracy of 58.8% highlighting that performing well on $\mathcal{ART}$ requires models to go far beyond the linguistic notion of entailment.

**Pre-trained Language Models**    BERT (Devlin et al., 2018) and GPT (Radford, 2018) have recently been shown to achieve state-of-the-art results on several NLP benchmarks (Wang et al., 2018). We fine-tune both BERT-Large and GPT as suggested in previous work and we present each instance in their natural narrative order. BERT-ft (fully connected) is the best performing model achieving 68.9% accuracy, compared to GPT's 63.1%.[8] Our AF approach was able to reduce BERT performance from over 88% by 20 points.

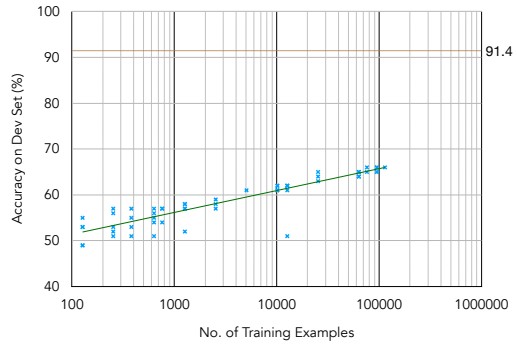

Figure 5: BERT learning curve on the *dev* set of $\mathcal{ART}$. For each point on the x-axis, we fine-tune BERT with five random seeds. Human performance is 91.4%.

**Learning Curve and Dataset Size**    While there is enough scope for considerably scaling up the dataset based on ROCStories, the learning curve in Figure 5 shows that the performance of the best model plateaus after $\sim$10,000 instances. In addition, there is still a wide gap ($\sim$23%) between the performance of the best model and human performance.

---

[7] Additional crowdsourcing details in the Appendix A.1

[8] The input format for the GPT model and BERT variants is described in the Appendix A.4.

| Model | BLEU | METEOR | ROUGE | CIDEr | BERT-Score | Human |
|---|---|---|---|---|---|---|
| GPT2-Fixed | 0.0 | 9.29 | 9.99 | 3.34 | 36.69 | - |
| $O_1$-$O_2$-Only | 2.23 | 16.71 | 22.83 | **33.54** | **48.74** | 42.26 |
| COMeT-Txt+GPT2 | 2.29 | 16.73 | 22.51 | 31.99 | 48.46 | 38.28 |
| COMeT-Emb+GPT2 | **3.03** | **17.66** | **22.93** | 32.00 | 48.52 | **44.56** |
| Human-written Hypotheses | 8.25 | 26.71 | 30.40 | 53.56 | 53.30 | 96.03 |

Table 2: Performance of generative models on the *test* set of $\mathcal{ART}$. All models except GPT2-Fixed are finetuned on $\mathcal{ART}$.

**GPT Adversary** Table 1 also includes results of our experiments where GPT was used as the adversary. Notably, in this case, adversarially filtering the dataset brings down GPT performance under 53%. On the other hand, the best BERT model, that encodes the *fully connected* bayesian network performs significantly better than the BERT model that encodes the *linear chain* assumptions – 72% compared to 65%. Therefore, we use the BERT fully connected model as the adversary in $\mathcal{ART}$. The gap between the linear chain and fully connected BERT models diminishes when BERT is used as an adversary – in spite of being a more powerful model – which indicates that adversarial filtering disproportionately impacts the model used as the adversary. However, the dataset also becomes more difficult for the other models that were not used as adversaries. For example, before any filtering, BERT scores 88% and OpenGPT gets 80%, which is much higher than either model achieves in Table 1 when the other model is used for filtering. This result is a reasonable indicator, albeit not a guarantee, that $\mathcal{ART}$ will remain challenging for new models released in the future.

## 5.2 Abductive Natural Language Generation

**Generative Language Models** As described in Equation 4, we train GPT2 conditioned on the tokens of the two observations $O_1$ and $O_2$. Both observations are enclosed with field-specific tags. ATOMIC (Sap et al., 2019), a repository of inferential *if-then* knowledge is a natural source of background commonsense required to reason about narrative contexts in $\mathcal{ART}$. Yet, there is no straightforward way to include such knowledge into a neural model as ATOMIC's nodes are not canonicalized and are represented as short phrases of text. Thus, we rely on COMeT – a transformer model trained on ATOMIC that generates nine commonsense inferences of events in natural language.[9] Specifically, we experiment with two ways of integrating information from COMeT in GPT2: (i) *as textual phrases*, and (ii) *as embeddings*.

Figure 3 shows how we integrate COMeT representations. Concretely, after the input tokens are embedded by the word-embedding layer, we append eighteen (corresponding to nine relations for each observation) embeddings to the sequence before passing through the layers of the Transformer architecture. This allows the model to learn each token's representation while attending to the COMeT embeddings – effectively integrating background commonsense knowledge into a language model.[10]

**Discussion** Table 2 reports results on the $\alpha$NLG task. Among automatic metrics, we report BLEU-4 (Papineni et al., 2002), METEOR (Banerjee & Lavie, 2005), ROUGE (Lin, 2004), CIDEr (Vedantam et al., 2015) and BERT-Score (Zhang et al., 2019) (with the `bert-base-uncased` model). We establish human performance through crowdsourcing on AMT. Crowdworkers are shown pairs of observations and a generated hypothesis and asked to label whether the hypothesis explains the given observations. The last column reports the human evaluation score. The last row reports the score of a held-out human-written hypothesis and serves as a ceiling for model performance. Human-written hypotheses are found to be correct for 96% of instances, while our best generative models, even when enhanced with background commonsense knowledge, only achieve 45% – indicating that the $\alpha$NLG generation task is especially challenging for current state-of-the-art text generators.

---

[9] Please see Appendix A.6 for a full list of the nine relations.

[10] We describe the format of input for each model in Appendix A.7.

# 6 ANALYSIS

## 6.1 $\alpha$NLI

**Commonsense reasoning categories** We investigate the categories of commonsense-based abductive reasoning that are challenging for current systems and the ones where the best model over-performs. While there have been previous attempts to categorize commonsense knowledge required for entailment (LoBue & Yates, 2011; Clark et al., 2007), crowdsourcing this task at scale with high fidelity and high agreement across annotators remains challenging. Instead, we aim to probe the model with soft categories identified by matching lists of category-specific keywords to the hypothesis choices.

| Category | Human Accuracy | BERT Accuracy | $\Delta$ |
|---|---|---|---|
| All $(1,000)$ | 91.4 | 68.8 | 22.6 |
| Numerical $(44)$ | 88.6 | 56.8 | 21.8 |
| Spatial $(130)$ | 91.5 | 65.4 | 26.1 |
| Emotional $(84)$ | 86.9 | 72.6 | 14.3 |

Table 3: BERT's performance and human evaluation on categories for 1,000 instances from the test set, based on commonsense reasoning domains (Numerical, Spatial, Emotional). The number in parenthesis indicates the size of the category.

Table 3 shows the accuracy of the best model (BERT-ft) across various categories of commonsense knowledge. BERT-ft significantly underperforms on instances involving *Numerical* ($56.8\%$) and *Spatial* ($65.4\%$) commonsense. These two categories include reasoning about numerical quantities and the spatial location of agents and objects, and highlight some of the limitations of the language models. In contrast, it significantly overperforms on the *Emotional* category ($72.6\%$) where the hypotheses exhibit strong textual cues about emotions and sentiments.

**Implausible transitions** A model for an instance of the $\mathcal{ART}$ dataset should discard implausible hypotheses in the context of the two given observations. In narrative contexts, there are three main reasons for an implausible hypothesis to be labeled as such:

| Story Transition | % of Dataset | BERT-ft Fully Connected Acc. (%) | BERT-ft Linear Chain Acc. (%) |
|---|---|---|---|
| $O_1 \nrightarrow h^-$ | 32.5 | 73.6 | 71.6 |
| $h^- \nrightarrow O_2$ | 45.3 | 69.0 | 70.5 |
| Plausible | 22.2 | 62.5 | 58.5 |
| All (1,000) | 100.0 | 69.1 | 68.2 |

Table 4: Fraction of dataset for which a particular transition in the story is broken for the negative hypothesis, for 1,000 random instances from the test set.

1. $O_1 \nrightarrow h^-$: $h^-$ is unlikely to follow after the first observation $O_1$.
2. $h^- \nrightarrow O_2$: $h^-$ is plausible after $O_1$ but unlikely to precede the second observation $O_2$.
3. Plausible: $\langle O_1, h^-, O_2 \rangle$ is a coherent narrative and forms a plausible alternative, but it is less plausible than $\langle O_1, h^+, O_2 \rangle$.

We analyze the prevalence of each of these reasons in $\mathcal{ART}$. We design a crowdsourcing task in which we show the implausible option along with the narrative context $\langle O_1, O_2 \rangle$ and get labels for which transition ($O_1 \nrightarrow h^-$, $h^- \nrightarrow O_2$ or neither) in the narrative chain is broken. Table 4 shows the proportion of each category from a subset of 1,000 instances from the *test* set. While $h^- \nrightarrow O_2$ accounts for almost half of the implausible transitions in $\mathcal{ART}$, all three categories are substantially present in the dataset. BERT performance on each of these categories indicates that the model finds it particularly hard when the narrative created by the incorrect hypothesis is plausible, but less plausible than the correct hypothesis. On that subset of the test set, the fully connected model performs better than the linear chain model where it is important to consider both observations jointly to arrive at the more likely hypothesis.

## 6.2 $\alpha$NLG

Figure 6 shows some examples of generations from the trained models compared to human-written generations. The example on the left is an example of an instance that only humans could get correct, while for the one on the right, `COMeT-Emb+GPT2` also generates the correct explanation for the observations.

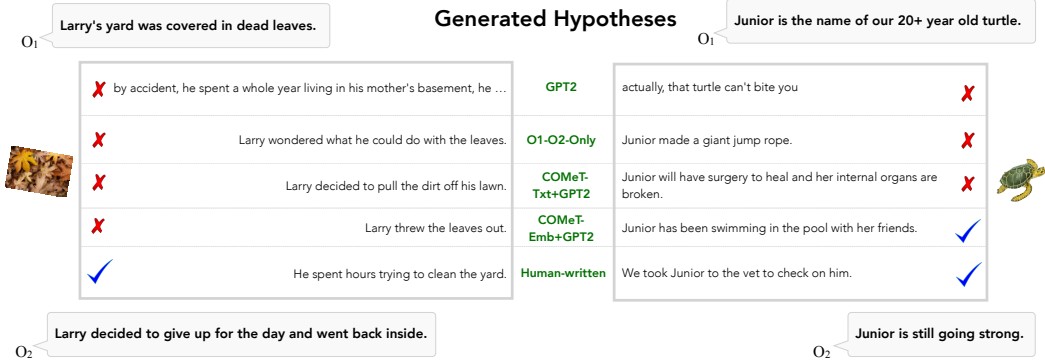

Figure 6: Examples of generated hypotheses from different models and human-written hypothesis for 2 instances from $\mathcal{ART}$.

## 7 TRANSFER LEARNING FROM $\mathcal{ART}$

$\mathcal{ART}$ contains a large number of questions for the novel abductive reasoning task. In addition to serving as a benchmark, we investigate if $\mathcal{ART}$ can be used as a resource to boost performance on other commonsense tasks. We apply transfer learning by first training a model on $\mathcal{ART}$, and subsequently training on four target datasets – WinoGrande Sakaguchi et al. (2020), WSC Levesque et al. (2011), DPR Rahman & Ng (2012) and HellaSwag Zellers et al. (2019). We show that compared to a model that is only trained on the target dataset, a model that is sequentially trained on $\mathcal{ART}$ first and then on the target dataset can perform better. In particular, pre-training on $\mathcal{ART}$ consistently improves performance on related datasets when they have relatively few training examples.

On the other hand, for target datasets with large amounts of training data, pre-training on $\mathcal{ART}$ does not provide a significant improvement.

## 8 RELATED WORK

**Cloze-Style Task vs. Abductive Reasoning** Since abduction is fundamentally concerned with *plausible* chains of cause-and-effect, our work draws inspiration from previous works that deal with narratives such as script learning

| Dataset | BERT-ft(D) | BERT-ft($\mathcal{ART}$)→ BERT-ft(D) |
|---|---|---|
| WinoGrande Sakaguchi et al. (2020) | 65.8% | **67.2%** |
| WSC Levesque et al. (2011) | 70.0% | **74.0%** |
| DPR Rahman & Ng (2012) | 72.5% | **86.0%** |
| Hellaswag Zellers et al. (2019) | **46.7%** | 46.1% |

Table 5: Transfer Learning from $\mathcal{ART}$

(Schank & Abelson, 1975) and the narrative cloze test (Chambers & Jurafsky, 2009; Jans et al., 2012; Pichotta & Mooney, 2014; Rudinger et al., 2015). Rather than learning prototypical scripts or narrative chains, we instead reason about the most plausible events conditioned on observations. We make use of the ROCStories dataset (Mostafazadeh et al., 2016), which was specifically designed for the narrative cloze task. But, instead of reasoning about plausible event sequences, our task requires reasoning about plausible explanations for narrative omissions.

**Entailment vs. Abductive Reasoning** The formulation of $\alpha$NLI is closely related to entailment NLI, but there are two critical distinctions that make abductive reasoning uniquely challenging. First, abduction requires reasoning about commonsense implications of observations (e.g., if we observe that the "grass is wet", a likely hypothesis is that "it rained earlier") which go beyond the linguistic notion of entailment (also noted by Josephson (2000)). Second, abduction requires non-monotonic reasoning about a set of commonsense implications collectively, to check the potential contradictions against multiple observations and to compare the level of plausibility of different hypotheses. This makes abductive reasoning distinctly challenging compared to other forms of reasoning such as induction and deduction (Shank, 1998). Perhaps more importantly, abduction is closely related to the kind of reasoning humans perform in everyday situations, where information is incomplete and definite inferences cannot be made.

**Generative Language Modeling**   Recent advancements in the development of large-scale pre-trained language models (Radford, 2018; Devlin et al., 2018; Radford et al., 2019) have improved the quality and coherence of generated language. Although these models have shown to generate reasonably coherent text when condition on a sequence of text, our experiments highlight the limitations of these models to 1) generate language non-monotonically and 2) adhere to commonsense knowledge. We attempt to overcome these limitations with the incorporation of a generative commonsense model during hypothesis generation.

**Related Datasets**   Our new resource $\mathcal{ART}$ complements ongoing efforts in building resources for natural language inference (Dagan et al., 2006; MacCartney & Manning, 2009; Bowman et al., 2015; Williams et al., 2018a; Camburu et al., 2018). Existing datasets have mostly focused on textual entailment in a deductive reasoning set-up (Bowman et al., 2015; Williams et al., 2018a) and making inferences about plausible events (Maslan et al., 2015; Zhang et al., 2017). In their typical setting, these datasets require a system to *deduce* the logically entailed consequences of a given *premise*. In contrast, the nature of abduction requires the use of commonsense reasoning capabilities, with less focus on lexical entailment. While abductive reasoning has been applied to entailment datasets (Raina et al., 2005), they have been applied in a logical theorem-proving framework as an intermediate step to perform textual entailment – a fundamentally different task than $\alpha$NLI.

## 9   CONCLUSION

We present the first study that investigates the viability of language-based abductive reasoning. We conceptualize and introduce Abductive Natural Language Inference ($\alpha$NLI) – a novel task focused on abductive reasoning in narrative contexts. The task is formulated as a multiple-choice question-answering problem. We also introduce Abductive Natural Language Generation ($\alpha$NLG) – a novel task that requires machines to generate plausible hypotheses for given observations. To support these tasks, we create and introduce a new challenge dataset, $\mathcal{ART}$, which consists of 20,000 commonsense narratives accompanied with over 200,000 explanatory hypotheses. In our experiments, we establish comprehensive baseline performance on this new task based on state-of-the-art NLI and language models, which leads to 68.9% accuracy with a considerable gap with human performance (91.4%). The $\alpha$NLG task is significantly harder – while humans can write a valid explanation 96% of times, the best generator models can only achieve 45%. Our analysis leads to new insights into the types of reasoning that deep pre-trained language models fail to perform – despite their strong performance on the closely related but different task of entailment NLI – pointing to interesting avenues for future research. We hope that $\mathcal{ART}$ will serve as a challenging benchmark for future research in language-based abductive reasoning and the $\alpha$NLI and $\alpha$NLG tasks will encourage representation learning that enables complex reasoning capabilities in AI systems.

## ACKNOWLEDGMENTS

We thank the anonymous reviewers for their insightful feedback. This research was supported in part by NSF (IIS-1524371), the National Science Foundation Graduate Research Fellowship under Grant No. DGE 1256082, DARPA CwC through ARO (W911NF15-1- 0543), DARPA MCS program through NIWC Pacific (N66001-19-2-4031), and the Allen Institute for AI. Computations on `beaker.org` were supported in part by credits from Google Cloud.

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

## A    APPENDICES

### A.1    DATA COLLECTION DETAILS

We describe the crowdsourcing details of our data collection method.

**Task 1 - Plausible Hypothesis Options**    In this task, participants were presented an incomplete three-part story, which consisted of the first observation ($O_1$) and the second observation ($O_2$) of the story. They were then asked to complete the story by writing a probable middle sentence that explains why the second observation should follow after the first one. We instructed participants to make sure that the plausible middle sentence (1) is short (fewer than 10 words) and (2) simple as if narrating to a child, (3) avoids introducing any extraneous information, and (4) uses names instead of pronouns (e.g., he/she) wherever possible.

All participants were required to meet the following qualification requirements: (1) their location is in the US, (2) HIT approval rate is greater than 95(%), and (3) Number of HITs approved is greater than 5,000. The reward of this task was set to be $0.07 per question ($14/hour in average), and each HIT was assigned to five different workers (i.e., 5-way redundancy).

**Task 2 - Implausible Hypothesis Options**    In this task, participants were presented a three-part story, which consisted of the first observation ($O_1$), a middle sentence ($h^+$) collected in Task 1, and the second observation ($O_2$) of the story. They were then asked to rewrite the middle sentence ($h^+$) with **minimal changes**, so that the story becomes *unlikely, implausible* or *inconsistent* ($h^-$). We asked participants to add or remove at most four words to $h^+$, while ensuring that the new middle sentence is grammatical. In addition, we asked them to stick to the context in the given story. For example, if the story talks about "doctors", they are welcome to talk about "health" or "diagnosis", but not mention "aliens". Finally, we also asked workers to verify if the given middle ($h^+$) makes a plausible story, in order to confirm the plausibility of $h^+$ collected in Task 1.

With respect to this task's qualification, participants were required to fulfill the following requirements: (1) their location is the US or Canada, (2) HIT approval rate is greater than or equal to 99(%), and (3) number of HITs approved is greater than or equal to $10,000$. Participants were paid $0.1 per question ($14/hour in average), and each HIT was assigned to three different participants (i.e., 3-way redundancy).

**Task 3 - $\alpha$NLI Human Performance**    Human performance was evaluated by asking participants to answer the $\alpha$NLI questions. Given a narrative context $\langle O_1, O_2 \rangle$ and two hypotheses, they were asked to choose the more plausible hypothesis. They were also allowed to choose "None of the above" when neither hypothesis was deemed plausible.

We asked each question to seven participants with the following qualification requirements: (1) their location is either in the US, UK, or Canada, (2) HIT approval rate is greater than 98(%), (3) Number of HITs approved is greater than $10,000$. The reward was set to $0.05 per HIT. We took the majority vote among the seven participants for every question to compute human performance.

### A.2    $\mathcal{ART}$ DATA STATISTICS

Table 6 shows some statistics of the $\mathcal{ART}$ dataset.

### A.3    FINE-TUNING BERT

We fine-tuned the BERT model using a grid search with the following set of hyper-parameters:

- batch size: $\{3, 4, 8\}$
- number of epochs: $\{3, 4, 10\}$
- learning rate: $\{$1e-5, 2e-5, 3e-5, 5e-5$\}$

The warmup proportion was set to $0.2$, and cross-entropy was used for computing the loss. The best performance was obtained with a batch size of $4$, learning rate of 5e-5, and number of epochs equal to 10. Table 7 describes the input format for GPT and BERT (and its variants).

|  | Train | Dev | Test |
|---|---|---|---|
| *Total unique occurrences* | | | |
| Contexts $\langle O_1, O_2 \rangle$ | 17,801 | 1,532 | 3,059 |
| Plausible hyp. $h^+$ | 72,046 | 1,532 | 3,059 |
| Implausible hyp. $h^-$ | 166,820 | 1,532 | 3,059 |
| *Avg. size per context* | | | |
| Plausible hyp. $h^+$ | 4.05 | 1 | 1 |
| Implausible hyp. $h^-$ | 9.37 | 1 | 1 |
| *Avg. word length* | | | |
| Plausible hyp. $h^+$ | 8.34 | 8.62 | 8.54 |
| Implausible hyp. $h^-$ | 8.28 | 8.55 | 8.53 |
| First observation $O_1$ | 8.09 | 8.07 | 8.17 |
| Second observation $O_2$ | 9.29 | 9.3 | 9.31 |

Table 6: Some statistics summarizing the $\mathcal{ART}$ dataset. The *train* set includes all plausible and implausible hypotheses collected via crowdsourcing, while the *dev* and *test* sets include the hypotheses selected through the Adversarial Filtering algorithm.

## A.4   BASELINES

The SVM classifier is trained on simple features like word length, overlap and sentiment features to select one of the two hypothesis choices. The bag-of-words baseline computes the average of GloVe (Pennington et al., 2014) embeddings for words in each sentence to form sentence embeddings. The sentence embeddings in a story (two observations and a hypothesis option) are concatenated and passed through fully-connected layers to produce a score for each hypothesis. The accuracies of both baselines are close to 50% (SVM: 50.6; BOW: 50.5).

Specifically, we train an SVM classifier and a bag-of-words model using GLoVE embeddings. Both models achieve accuracies close to 50%. An Infersent (Conneau et al., 2017) baseline that uses sentences embedded by max-pooling over Bi-LSTM token representations achieves only 50.8% accuracy.

| Model | Input Format |
|---|---|
| GPT | `[START]` $O_1$ + $h^i$ `[SEP]` $O_2$ `[SEP]` |
| BERT-ft [Hypothesis Only] | `[CLS]` $h^i$ `[SEP]` |
| BERT-ft [First Observation Only] | `[CLS]` $O_1$ `[SEP]` $h^i$ `[SEP]` |
| BERT-ft [Second Observation Only] | `[CLS]` $h^i$ `[SEP]` $O_2$ `[SEP]` |
| BERT-ft [Linear Chain] | `[CLS]` $O_1$ `[SEP]` $h^i$ `[SEP]` ; `[CLS]` $h^i$ `[SEP]` $O_2$ `[SEP]` |
| BERT-ft [Fully Connected] | `[CLS]` $O_1$ + $O_2$ `[SEP]` $h^i$ `[SEP]` |

Table 7: Input formats for GPT and BERT fine-tuning.

## A.5   ADVERSARIAL FILTERING OF HYPOTHESES CHOICES

Given an observation pair and sets of plausible and implausible hypotheses $\langle O_1, O_2, \mathcal{H}^+, \mathcal{H}^- \rangle$, our adversarial filtering algorithm selects one plausible and one implausible hypothesis $\langle O_1, O_2, h^+, h^- \rangle$ such that $h^+$ and $h^-$ are hard to distinguish between. We make three key improvements over the previously proposed Adversarial Filtering (AF) approach in Zellers et al. (2018). First, Instead of a single positive sample, we exploit a pool $\mathcal{H}^+$ of *positive samples* to choose from (i.e. plausible hypotheses). Second, Instead of machine generated distractors, the pool $\mathcal{H}^-$ of *negative samples* (i.e. implausible hypotheses) is human-generated. Thus, the distractors share stylistic features of the positive samples as well as that of the context (i.e. observations $O_1$ and $O_2$) – making the negative samples harder to distinguish from positive samples. Finally, We use BERT (Devlin et al., 2018) as

the adversary and introduce a *temperature* parameter that controls the maximum number of instances that can be modified in each iteration of AF. In later iterations, fewer instances get modified resulting in a smoother convergence of the AF algorithm (described in more detail below).

Algorithm 1 provides a formal description of our approach. In each iteration $i$, we train an adversarial model $M_i$ on a random subset $\mathcal{T}_i$ of the data and update the validation set $\mathcal{V}_i$ to make it more challenging for $M_i$. For a pair $(h_k^+, h_k^-)$ of plausible and implausible hypotheses for an instance $k$, we denote $\delta = \Delta_{M_i}(h_k^+, h_k^-)$ the difference in the model evaluation of $h_k^+$ and $h_k^-$. A positive value of $\delta$ indicates that the model $M_i$ favors the plausible hypothesis $h_k^+$ over the implausible one $h_k^-$. With probability $t_i$, we update instance $k$ that $M_i$ gets correct with a pair $(h^+, h^-) \in \mathcal{H}_k^+ \times \mathcal{H}_k^-$ of hypotheses that reduces the value of $\delta$, where $\mathcal{H}_k^+$ (resp. $\mathcal{H}_k^-$) is the pool of plausible (resp. implausible) hypotheses for instance $k$ .

We ran AF for 50 iterations and the temperature $t_i$ follows a sigmoid function, parameterized by the iteration number, between $t_s = 1.0$ and $t_e = 0.2$. Our final dataset, $\mathcal{ART}$, is generated using BERT as the adversary in Algorithm 1.

---

**Algorithm 1:** Dual Adversarial Filtering

**input** : dataset $\mathcal{D}_0$, plausible & implausible hypothesis sets $(\mathcal{H}^+, \mathcal{H}^-)$, number of iterations $n$, initial & final temperatures $(t_s, t_e)$

**output:** dataset $\mathcal{D}_n$

1 **for** *iteration* $i : 0..n-1$ **do**
2      $t_i = t_e + \frac{t_s - t_e}{1 + e^{0.3(i - \frac{3n}{4})}}$
3      Randomly partition $\mathcal{D}_i$ into $(\mathcal{T}_i, \mathcal{V}_i)$.
4      Train model $M_i$ on $\mathcal{T}_i$.
5      $\mathcal{S}_i = \emptyset$, the selected hypotheses for $\mathcal{V}_i$.
6      **for** $(h_k^+, h_k^-) \in \mathcal{V}_i$ **do**
7          Pick $r$ uniformly at random in $[0, 1]$.
8          **if** $r > t_i$ or $\Delta_{M_i}(h_k^+, h_k^-) < 0$ **then**
9              Add $(h_k^+, h_k^-)$ to $\mathcal{S}_i$.
10          **else**
11              Pick $(h^+, h^-) \in \mathcal{H}_k^+ \times \mathcal{H}_k^-$ s.t. $\Delta_{M_i}(h^+, h^-) < \Delta_{M_i}(h_k^+, h_k^-)$
12              Add $(h^+, h^-)$ to $\mathcal{S}_i$.
13          **end**
14      **end**
15      $\mathcal{D}_{i+1} = \mathcal{T}_i \cup \mathcal{S}_i$
16 **end**

---

### A.6 ATOMIC RELATIONS

ATOMIC (Sap et al., 2019) represents commonsense knowledge as a graph with events are nodes and the following nine relations as edges:

1. xIntent: Why does X cause an event?

2. xNeed: What does X need to do before the event?

3. xAttr: How would X be described?

4. xEffect: What effects does the event have on X?

5. xWant: What would X likely want to do after the event?

6. xReaction: How does X feel after the event?

7. oReact: How do others' feel after the event?

8. oWant: What would others likely want to do after the event?

9. oEffect: What effects does the event have on others?

| Model | Input Format |
|---|---|
| GPT2-Fixed | $w_1^1 \ldots w_n^1 w_1^2 \ldots w_n^2$ Because, |
| $O_1$-$O_2$-Only | $\langle o1 \rangle w_1^1 \ldots w_n^1 \langle /o1 \rangle \langle o2 \rangle w_1^2 \ldots w_n^2 \langle /o2 \rangle \langle h \rangle$ |
| COMeT-Txt+GPT2 | $\langle p_1^1 \rangle T_1^1 \ldots T_9^1 \langle p_9^1 \rangle \langle p_1^2 \rangle T_1^2 \ldots T_9^2 \langle p_9^2 \rangle \langle o1 \rangle w_1^1 \ldots w_n^1 \langle /o1 \rangle \langle o2 \rangle w_1^2 \ldots w_n^2 \langle /o2 \rangle \langle h \rangle$ |
| COMeT-Emb+GPT2 | $c_1^1 \ldots c_9^1; c_1^2 \ldots c_9^2 \langle o1 \rangle w_1^1 \ldots w_n^1 \langle /o1 \rangle \langle o2 \rangle w_1^2 \ldots w_n^2 \langle /o2 \rangle \langle h \rangle$ |

Table 8: Input format used to training and generated text from various GPT2 based models. $c_i^j$ refers to the COMeTembeddings obtained using a separate transformer model for relation $i$ and observation $j$. Similarly, $T_i^j$ is the textual phrase for relation $i$, observation $j$. Where appropriate, field specific start and end-tags are added to the sequence of inputs.

## A.7 GENERATION MODELS INPUT FORMAT

Table 8 describes the format of input to each variation of the generative model evaluated.

