# OpenReview forum: "Abductive Commonsense Reasoning"
_ICLR.cc/2020/Conference — Accept (Poster)_

### Official Review · AnonReviewer3 · 2019-10-22
**Official Blind Review #3**

**Rating:** 6

**Review:**

Summary: the paper purposes a dataset of abductive language inference and generation. The dataset is generated by human, while the testing set is adversarially selected using BERT. The paper experiments the popular deep learning models on the dataset and observe shortcoming of deep learning on this task.

Comments: overall, the problem on abductive inference and abductive generation in language in very interesting and important. This dataset seems valuable. And the paper is simple and well-written.

Concerns: I find the claim on deep networks kind of irresponsible.
1. The dataset is adversarially filtered using BERT and GPT, which gives deep learning model a huge disadvantage. After all, the paper says BERT scores 88% before the dataset is attacked.
2. The human score of 91.4% is based on majority vote, which should be compared with an ensemble of deep learning prediction. To compare the author should use the average score of human.
3. The ground truth is selected by human.

On a high level, the main difficulty of abduction is to search in the exponentially large space of hypothesis. Formulating the abduction task as a (binary) classification problem is less interesting. The generative task is a better option.

Decision: despite the seeming unfair comparison, this task is novel. I vote for weak accept.

**Experience Assessment:**

I have published one or two papers in this area.

**Review Assessment: Checking Correctness Of Derivations And Theory:**

I assessed the sensibility of the derivations and theory.

**Review Assessment: Checking Correctness Of Experiments:**

I assessed the sensibility of the experiments.

**Review Assessment: Thoroughness In Paper Reading:**

I read the paper thoroughly.

---

> ### Author Response · Authors · 2019-11-14
> **Thank you for the positive feedback!**
>
> We appreciate AnonReviewer3 for encouraging comments about the importance of the proposed abductive inference and generation tasks and about the value of our proposed dataset. We address the main concerns individually below:
>
>
>
> Adversarially filtering using BERT and GPT gives deep learning models a disadvantage:
> While BERT originally achieved high performance on the originally collected dataset, several recent studies [1][2][3][4] have found the presence of annotation artifacts in crowdsourced data that inadvertently leak information about the target label. This subsequently leads to overestimation of the performance of AI systems on end tasks. Our adversarial filtering (AF) algorithm aims to address the problem of overestimation of performance. In spite of targeting GPT/BERT during AF, human performance on the AF resulting dataset is still high. The significant gap between human and BERT performance leaves scope for inventing new methods for abductive reasoning.
>
>
>
> Ensemble of BERT models:
> An ensemble of three BERT models achieves an accuracy of 68.9%, very close to a single model 68.6%.
>
>
> Average score of human:
> The average score of human annotations is 89.4%. This is directly comparable with BERT-Ft [Fully Connected] model’s performance of 68.6% in Table 1.
>
>
> Re. Ground Truth:
> The ground truth is assigned based on whether a hypothesis was collected during the plausible (Appendix A1 Task1) or implausible (Appendix A1 Task2) phase of the data collection procedure. To measure human performance, we had three annotators select the correct hypothesis and measured human performance as the accuracy of their majority-vote. Please let us know if this answers your question. If not, could you please clarify your question?
>
>
>
> Generative task vs classification:
> We completely agree. While the generative task is more general and much more interesting, the challenge of evaluating generations is significant, particularly for this task. This is due to the fact that there could be multiple distinct plausible explanations for a given pair of hypothesis.
>
> Consider the following example:
>
> O1: Kelly and her friend wanted to take a train to the city.
> O2: They had to wait for another one.
>
> Plausible explanations:
> 1. They read the timetable incorrectly and arrived at the station just after a train had left.
> 2. The train was full.
>
> Both explanations are plausible, and explain the observations, but automated evaluation metrics are not reliable enough to capture this phenomenon based on their reliance on surface level similarities. To simultaneously make progress on the novel abductive reasoning task and due to the ease of evaluation, we additionally introduce a discriminative version of the task. Nonetheless, we agree that in its most general form, there could be any number of observations and models should be required to generate explanatory hypotheses in natural language (alpha-NLG task).
>
>
>
> [1] Gururangan et al. Annotation artifacts in natural language inference data.
> [2] Poliak et al. Hypothesis only baselines in natural language inference.
> [3] Tsuchiya et al. Performance impact caused by hidden bias of training data for recognizing textual entailment.
> [4] Sakaguchi et al. WINOGRANDE: An Adversarial Winograd Schema Challenge at Scale

---

### Official Review · AnonReviewer2 · 2019-10-24
**Official Blind Review #2**

**Rating:** 8

**Review:**

This paper introduces two new natural language tasks in the area of commonsense reasoning: natural language abductive  inference and natural language abductive  generation. The paper also introduces a new dataset, ART, to support training and evaluating models for the introduced tasks. The paper describes the new language abductive tasks, contrasting it to the related, and recently established, natural language inference (entailment) task.  They go on to describe the construction of baseline models for these tasks. These models were primarily constructed to diagnose potential unwanted biases in the dataset (e.g., are the tasks partially solved by looking at parts of the input, do existing NLI models far well on the dataset, etc.), demonstrating a significant gap with respect to human performance.

The paper, and the dataset specifically, represent an important contribution to the area of natural language commonsense reasoning. It convincingly demonstrates that the proposed tasks, while highly related to natural language entailment, are not trivially addressed by existing state-of-the-art models. I expect that teaching models to perform well on this task can lead to improvements in other tasks, although empirical evidence of this hypothesis is currently absent from the paper.

Below are a set of more specific observations about the paper. Some of these comments aim to improve the presentation or content of the paper.

1. In Section “5.1 - Pre-trained Language Models” and attendant Table 1describe results of different baselines on the ART inference task. The results in the table confused me for quite some time, I’d appreciate some clarifications. With respect to the differences with columns 1 (GPT AF) and 2 (ART, also BERT AF) I would like the comparison to be made more clear. As far as I understand it, there are 2 parts of the dataset that can be varied: (1) the train+dev sets and (2) the test set. Furthermore, it seems that it makes sense to vary each of these at a time, if we are to compare results with variants. For example: we can fix the test set, and vary how we generate training and dev examples. If a model does better with the same test set, we can assume the train+dev examples were better for the model (for whatever reasons, closer distribution to test, harder or more
informative training examples, etc). We can also keep the train+set constant, and vary the test set. This allows us to evaluate which test set is harder with respect to the training examples. The caption of Table 1 implies that both columns are evaluations based on the “ART” test set. If that is correct, then the train+dev set generated from the GPT adversarial examples is of better quality, generating a BERT-ft (fully connected) model that is 3% better. But the overall analysis seems to indicate that this is not what was done in the experiment. Rather, it seems that *both* the train+dev _and_ the test sets were modified concurrently. If that is the case, I would emphasize that the text needs to make this distinction clear. Furthermore, I would say that varying both train and test sets concurrently is sub-optimal, and makes it a bit harder to draw the conclusion that BERT adversarial filtering leads to a stronger overall dataset.

2. Along the lines of the argument in (1), above, I would urge the authors to publish the *entire* set of generated hypotheses (plausible and implausible) instead of only releasing the adversarially filtered pairs. Our group’s experience with training inference models is that it is often beneficial to train using “easy” examples, not only hard examples. I suspect the adversarially filtered set will focus on hard examples only. While this is fine to do in the test set, I think if the full set of annotated/generated hypotheses are released, model designers can experiment with combining pairs of hypothesis in different ways.

3. Furthering the argument of (2): in certain few-shot classification tasks, one is typically asked to identify similarity between one test example and different class representatives. Experience shows that it is often beneficial to train the model on a larger number of distractor classes than what the model is eventually evaluated on (e.g., https://papers.nips.cc/paper/6996-prototypical-networks-for-few-shot-learning). In the alpha-NLI setting, have you experimented with training using multiple distractors, instead of only 1, during training (even if you end up evaluating over 2 hypotheses)?

4. One potential argument for introducing a new natural language task is of transfer learning: learning to perform a complex natural language task should lead to better natural language models more generally, or, for some other related tasks. This paper does not really touch on this aspect of the work. But, potentially, one investigation that could be conducted is through a reversal of the paper’s existing NLI entailment experiment. The paper shows that NLI entailment models do not perform well on alpha-NLI. But it would be interesting to see if a model trained on alpha-NLI, and fine-tuned or multi-tasked on NLI entailment, does better on NLI entailment (i.e., is there transfer from alpha-NLI to entailment NLI?).

5. Another option is to evaluate whether alpha-NLI helps with other commonsense tasks. One other example is Winograd Schema Challenge, which current systems also perform well below human performance. It also seems that the Winograd Schema Challenge questions are not too far from abductive inference.

6. In the abstract of the paper, before the paper defines the abductive inference/generation task specifically, the claim that abductive reasoning has had much attention in research seemed awkward. Informally, most commonsense reasoning (including NLI entailment) could be cast as abductive reasoning.

7. In at least one occasion, I found an acronym which was hard to find the definition for (“AF” used in Section “5.1 - Pre-trained Language Models”; I assumed it was “adversarial filtering”.)

8. In Section “5.1 - Pre-trained Language Models” it seems that the text quotes an accuracy for BERT-ft (fully connected) of 68.9%, but Table 1 indicates 69.6%.

9. In Section “5.1 - Learning Curve and Dataset Size”, there is a claim that the performance of the model plateaus at ~10,000 instances. This does not seem supported by Figure 5. There appears to be over 5-7% accuracy (absolute) improvements from 10k to 100k examples. Maybe the graph needs to be enhanced for legibility?

10. It is great that the paper includes human numbers for both tasks, including all the metrics for generation.

11. Period missing in footnote 8.

12. The analysis section is interesting, it is useful to have in the paper. However, Table 3 is a bit disappointing in that ~26% of the sampled examples fit into one of the categories. It would be great if the authors could comment on the remaining ~74% of the sampled dataset.


**Experience Assessment:**

I have read many papers in this area.

**Review Assessment: Checking Correctness Of Derivations And Theory:**

I carefully checked the derivations and theory.

**Review Assessment: Checking Correctness Of Experiments:**

I carefully checked the experiments.

**Review Assessment: Thoroughness In Paper Reading:**

I read the paper thoroughly.

---

> ### Author Response · Authors · 2019-11-14
> **Pre-training on alpha-NLI consistently improves performance on datasets that have relatively small training sets**
>
> We thank AnonReviewer2 for their appreciation of our work as an “important contribution” for commonsense reasoning and for their detailed review with several insightful suggestions. Based on these suggestions, we present new experimental results and address individual concerns below:
>
>
>
> Clarifications on Table 1 results:
> “GPT AF” column in Table 1 shows the performance of various models on a dataset that is generated by using GPT as the adversary. Similarly, the “ART” column uses BERT as the adversary. The aim of our adversarial filtering step is to arrive at a dataset that is hard for current state-of-the-art models. The table shows that using a strong model as the adversary results in a dataset that is hard for all other weaker models. Nonetheless, human performance remains high.
> At the end of our adversarial filtering algorithm, we have a large set of instances whose hypothesis choices are adversarially selected. This set of instances is randomly split into train/dev/test splits. While modifying only the train (or only the test) set is possible, the distribution of instances in the training and test splits will be distinct — thus making the results hard to interpret.
> We will update the text to make this distinction between col 1 and col 2 of Table 1 clear.
>
>
>
> Release the entire set of hypotheses:
> Yes, we will release all correct and incorrect hypotheses for the training instances, but the dev and test sets will only contain adversarially selected hypothesis pairs. Table 5 in the Appendix has the statistics of the ART dataset that we plan to make public with this work.
>
>
>
> Train with multiple distractors, test with one distractor:
> Thank you for the suggestion and the pointer to the paper. We performed this experiment and found that training with multiple distractors results in a performance of 68.1%, while training with a single distractor achieves 68.6%. While the performance difference is not significant, interestingly, training with multiple distractors seems to be more stable to hyperparameter changes (e.g. batch size, no. of epochs, learning rate). We will include these findings in the final version of the paper.
>
>
>
> Transfer learning to other tasks or language model:
> Thank you for another great suggestion. We have performed these experiments as well, and they show that pre-training on alpha-NLI consistently improves performance on datasets that have relatively few training examples.
> In particular, we compared the performance of 1) BERT fine-tuned on a given dataset and 2) BERT sequentially fine-tuned on alpha-NLI and then on the chosen dataset. For SNLI and HellaSwag, datasets with large amount of training data, there is no significant performance improvement due to pre-training on alpha-NLI.
>
>
> Dataset                            BERT-Ft(Dataset)                  BERT-Ft(alpha-NLI)→BERT-Ft(Dataset)
> --------------------------------------------------------------------------------------------------------------------------------
> WinoGrande [1]                       65.8%                                                               67.2%
> WSC[2]                                       70.0%                                                               74.0%
> DPR[3]                                       72.5%                                                                86.0%
> HellaSwag[4]                            46.7%                                                                46.1%
>
>
>
> Winograd Schema Challenge and NLI recast as Abductive Reasoning:
> While instances of WSC and NLI may require abductive reasoning, alpha-NLI is the first dataset to isolate this important, specific type of reasoning for future studies. Pre-training on alpha-NLI helps improve performance on WSC.
>
>
>
> Comment on the other commonsense categories:
> We considered more categories (e.g., negation, physical, temporal commonsense) but the performance difference between humans and BERT-ft in those categories was not statistically significant. We will include these categories in the camera ready. Nonetheless, we’d like to emphasize that reliable and consistent categorization of instances into commonsense categories is a challenge. Often, solving an instance might require multiple different types of commonsense reasoning and the distinction of different types of commonsense is also nebulous.
>
>
>
> Presentation concerns:
> Yes, AF meant adversarial filtering.
> The best performance of BERT-ft [Fully Connected] is 68.6%.
> We have updated the paper to show the plateauing effect more vividly (Please see Fig 6 in the updated paper).
> We have fixed the footnote in the updated PDF.
> We will fix the presentation issues in the final version.
>
>
> [1] Sakaguchi et al. WINOGRANDE: An Adversarial Winograd Schema Challenge at Scale
> [2] Levesque et. al. The winograd schema challenge.
> [3] Rahman and Ng. Resolving Complex Cases of Definite Pronouns: The Winograd Schema Challenge
> [4] Zellers et. al. HellaSwag: Can a Machine Really Finish Your Sentence?

---

### Official Review · AnonReviewer1 · 2019-10-24
**Official Blind Review #1**

**Rating:** 6

**Review:**

This paper proposes a new task/dataset for language-based abductive reasoning in narrative texts.

Pros:

-	The proposed task is interesting and well motivated. The paper contributes a dataset (20,000 commonsense narratives and 200,000 explanatory hypotheses). The construction of the dataset was performed carefully (e.g., avoiding annotation artifacts).

-	The paper established many reasonable baselines.

-	The paper conducted detailed analysis, which invites more research on this task: despite the strong performance of many existing systems on NLI/RTE, there are larger gaps between the performance of these models and human performance on the proposed task. The experiments well support the conclusions made in the paper.

-	The paper is well structured and easy to follow. It is well written.

Cons/comments:

-	While this is a new and interesting task, the contribution (as discussed above in “pros” above) is somewhat limited. I also suggest the paper discusses e-SNLI a bit more.

-	The paper has a specific form of formulation for abductive reasoning, where there are exactly two observations and one proceeds the other; the explanation happens in between. I can see this helps collect and annotate data, but also limit the form of abductive reasoning and how models should be developed.

-	Should the title of the paper specify the paper is about “language-based” abductive reasoning.

-	A minor one: “Table 7 reports results on the αNLI task.” Should it be “Table 2”?


**Experience Assessment:**

I have published in this field for several years.

**Review Assessment: Checking Correctness Of Derivations And Theory:**

I assessed the sensibility of the derivations and theory.

**Review Assessment: Checking Correctness Of Experiments:**

I assessed the sensibility of the experiments.

**Review Assessment: Thoroughness In Paper Reading:**

I read the paper at least twice and used my best judgement in assessing the paper.

---

> ### Author Response · Authors · 2019-11-14
> **Thank you for the positive feedback!**
>
> We thank AnonReviewer1 for their positive comments about the interesting-ness of our proposed abductive reasoning tasks (inference and generation) and the associated benchmark dataset. We address specific concerns individually below:
>
> Discussion about e-SNLI:
> A key distinction between e-SNLI and Abductive-NLI is that the explanations in e-SNLI serve the purpose of justifying model decisions. In contrast, the goal of Abductive-NLI and Abductive-NLG is to select or generate explanatory hypotheses for given observations. Indeed, analogous to e-SNLI for SNLI, Abductive-NLI can be extended to “e-Abductive-NLI” by providing explanations that justify the selected hypothesis.
> Consider the following example that BERT fails to predict correctly:
>
> O1: Chad loves Barry Bonds.
>     H1: Chad got to meet Barry Bonds online, chatting.
>     H2: Chad waited after a game and met Barry.
> O2: Chad ensured that he took a picture to remember the event.
>
> The e-Abductive-NLI task would require models to generate an explanation for selecting H2. For the above example, a possible explanation for selecting H2 could be: “People need to be physically co-located to take a picture with someone. Meeting online does not mean two people are physically co-located”.
> We think generating such justifications is a great next step and hope that our work will foster such interesting future research.
>
>
>
> Re. somewhat limited contribution:
> We appreciate the opportunity to briefly restate our contributions and to discuss its significance.
> Abductive Commonsense Reasoning, a critical capability in human reasoning, is relatively less studied in NLP research. To support this line of research, our work introduces a dataset that focuses explicitly on this important reasoning capability. Furthermore, several recent works [1,2,3,4] have shown the presence of annotation artifacts in crowdsourced datasets -- which poses a significant challenge for dataset curation. Our work makes the following contributions:
> i) proposes and formalizes two novel tasks of Abductive Inference and Abductive Generation,
> ii) presents a new dataset in support of these tasks collected through careful crowdsourcing design and an adversarial filtering algorithm,
> iii) establishes strong baselines on the task proving the difficulty of the tasks and
> iv) analyses the types of commonsense reasoning that current state of the art models fall short on.
>
>
>
> Re. limited form of Abductive Reasoning:
> The simplifying assumptions, mentioned in the paper, allow us to i) formulate the tasks concretely and ii) curate the dataset and evaluate models viably. We show that in spite of the assumptions, our dataset presents significant challenges for current models. We totally agree that in its most general form, there should be any number of observations and models should be required to generate explanatory hypotheses in natural language (as in the alpha-NLG task). We hope our work will lead to this future line of research.
>
>
>
> Re. the title:
> Thanks for the suggestion. We will update the title to reflect that this work is aimed at language-based abductive reasoning.
>
>
>
> Table 7 vs Table2:
> Thanks for catching that. We’ve updated the paper with the fix.
>
>
>
> [1] Gururangan et al. Annotation artifacts in natural language inference data.
> [2] Poliak et al. Hypothesis only baselines in natural language inference.
> [3] Tsuchiya e. al. Performance impact caused by hidden bias of training data for recognizing textual entailment.
> [4] Sakaguchi et al. WINOGRANDE: An Adversarial Winograd Schema Challenge at Scale

---

### Author Response · Authors · 2019-11-14
**Overall Comments (for all reviewers): Pre-training on alpha-NLI consistently improves performance on datasets that have small training sets**

We thank all reviewers for their overall positive comments about our work.

Based on AnonReviewer2's suggestion, we performed experiments that show that pre-training on alpha-NLI consistently improves performance on related datasets that have relatively few training examples.

In particular, we compared the performance of 1) BERT fine-tuned on a given dataset and 2) BERT sequentially fine-tuned on alpha-NLI and then on the chosen dataset.

Dataset                            BERT-Ft(Dataset)                  BERT-Ft(alpha-NLI)→BERT-Ft(Dataset)
--------------------------------------------------------------------------------------------------------------------------------
WinoGrande [1]                       65.8%                                                                67.2%
WSC[2]                                       70.0%                                                                74.0%
DPR[3]                                       72.5%                                                                86.0%
HellaSwag[4]                            46.7%                                                                46.1%

For SNLI and HellaSwag, datasets with large amount of training data, there is no significant performance improvement upon pre-training on alpha-NLI.

[1] Sakaguchi et al. WINOGRANDE: An Adversarial Winograd Schema Challenge at Scale
[2] Levesque et. al. The winograd schema challenge.
[3] Rahman and Ng. Resolving Complex Cases of Definite Pronouns: The Winograd Schema Challenge
[4] Zellers et. al. HellaSwag: Can a Machine Really Finish Your Sentence?

---

### Decision · Program_Chairs · 2019-12-19

**Decision:**

Accept (Poster)

**Comment:**

This paper presents a dataset, created using a combination of existing resources, crowdsourcing, and model-based filtering, that aims to tests models' understanding of typical progressions of events in everyday situations. The dataset represents a challenge for a range of state of the art models for NLP and commonsense reasoning, and also can be used productively as a training task in transfer learning.

After some discussion, reviewers came to a consensus that this represents an interesting contribution and a potentially valuable resource. There were some concerns—not fully resolved—about the implications of using model-based filtering during data creation, but these were not so serious as to invalidate the primary contributions of the paper.

While the thematic fit with ICLR is a bit weak—the primary contribution of the paper appears to be a dataset and task definition, rather than anything specific to representation learning—there are relevant secondary contributions, and I think that this work will be practically of interest to a reasonable fraction of the ICLR audience.